# Biomphalaria pfeifferi infected with Schistosoma spp. in Kakamega and Bungoma counties, western Kenya confirms autochthonous transmission of intestinal schistosomiasis

Maurice R. Odiere[1]*, Stella Kepha[2], Jimmy Kihara[2], Chitiavi Juma[3], Dickson Kioko[4], Florence Wakesho[4], Dollycate N. Wanja[3], Martin Muchangi[3], Ivy Sempele[5], Irene Chami[5], Sultani Hadley Matendechero[4], Wyckliff Omondi[4]

**1** Centre for Global Health Research, Kenya Medical Research Institute, Kisumu, Kenya, **2** Eastern and Southern Africa Centre of International Parasite Control, Kenya Medical Research Institute, Nairobi, Kenya, **3** Amref Health Africa, Nairobi, Kenya, **4** Division of Vector Borne and Neglected Tropical Diseases, Ministry of Health, Nairobi, Kenya, **5** The END Fund, New York, New York, United States of America

* Modiere@kemri.go.ke

## Abstract

### Background

Granular mapping conducted in Kakamega and Bungoma counties of western Kenya provided strong evidence of intestinal schistosomiasis among school-age children in this area. However, it was unclear whether the observed infections were due to active transmission. To address this gap, a follow-up malacology survey was conducted to determine the presence of snail vectors and ascertain active transmission.

### Methods

Nineteen Wards with ≥10% prevalence of *Schistosoma mansoni* were selected from Kakamega and Bungoma counties. From these Wards, 42 primary schools with the highest prevalence of *S. mansoni* were used to identify nearby water bodies with human-water contact activities for sampling snail vectors. Live snails were sampled by experienced collectors using a handheld dip-net scoop in standardized sampling frames for ~30 minutes at each water body. Snails were counted and identified to species-level based on shell morphology. Site locations sampled for snails were mapped using a geographical information system, and the type of water body recorded. All *Biomphalaria* and *Bulinus* snails were transported to the laboratory where they were screened for cercariae. Cercariae were identified to basic taxonomic groups using standard identification keys. The relative and mean abundance and prevalence of *Schistosoma* sp. infection in snails was determined.

**Data availability statement:** All data supporting the findings of this study are uploaded as supplementary file.

**Funding:** This work was supported by The END Fund through the Deworming Innovation Fund (DIF), Award Number: KEN-004-AMRK. The funders had no role in study design, data collection and analysis, decision to publish, or preparation of the manuscript.

**Competing interests:** The authors have declared that no competing interests exist.

## Results

Out of 4,245 snails collected, 3,341 (78.7%, 95% CI: 77.5-79.9) were putatively identified as *Biomphalaria pfeifferi*, 88 (2.1%, 95% CI: 1.7-2.5) as *Bulinus globosus*, 664 (15.6%, 95% CI: 14.6-16.7) as *Lymnaea natalensis* and 152 (3.6%) as other species. *B. pfeifferi* were found in 36 out of the 42 primary school regions (85.7%), with the highest number (605 snails) recorded at Indangalasia in Lusheya-Lubinu Ward. A total of 87 (2.6%, 95% CI: 2.07-3.14) *B. pfeifferi* shed *Schistosoma* sp. cercariae. The mean abundance of *B. pfeifferi* was similar between streams (16 ± 35) and swamps/marshes (20 ± 31) (P = 0.356), but was higher compared to rivers (4 ± 10) (P = 0.005).

## Conclusions

The high abundance of *B. pfeifferi* coupled with the observation of field-caught snails shedding cercariae confirms autochthonous schistosomiasis transmission corroborating the reported human infections in Kakamega and Bungoma counties. Incorporation of focal snail control to complement chemotherapy will accelerate interruption of transmission in these areas.

## Author summary

Recent mapping in Kakamega and Bungoma Counties of western Kenya showed many cases of intestinal schistosomiasis (bilharzia). However, it was unclear whether the transmission of the disease was taking place on site. To investigate this, a snail survey was conducted, because certain freshwater snails carry and release the parasite that infects people. The survey sampled water bodies near 42 primary school regions in 19 Wards where infection levels were high. Snails were collected, identified by species, and tested in the laboratory to see if they were releasing infectious parasite larvae. Out of 4,245 snails collected, most (about 79%) were *Biomphalaria pfeifferi*, the main carrier of *Schistosoma mansoni*. These snails were found at 36 out of 42 primary school regions sampled for snails, especially in streams and swamps. Importantly, 2.6% of them were actively shedding parasite larvae, meaning they were capable of infecting people. These findings confirm that schistosomiasis transmission is ongoing in these counties, not just a result of past exposure or imported cases. This study suggests that, in addition to treating people with medication, targeted snail control in high-risk areas could help stop transmission faster.

## Introduction

Schistosomiasis (SCH) is a snail-borne, inflammatory disease that infects an estimated 240 million people globally [1], with over 90% of the global burden borne by sub-Saharan Africa [2]. In Kenya, approximately 9 million people are estimated to be

infected [3] and approximately 17.4 million are at risk of schistosomiasis [4]. Both *Schistosoma mansoni* which is responsible for intestinal schistosomiasis and *S. haematobium*, which causes urogenital schistosomiasis are present in Kenya [3]. Considerable progress has been made in bringing schistosomiasis under control and with the aim of proceeding to elimination [5,6], with chemotherapy as the cornerstone of the intervention strategies. Although chemotherapy has considerably reduced prevalence and intensity of infection [7], it is acknowledged that mass drug administration (MDA) alone is insufficient to reduce transmission. The existence of persistent hotspots of schistosomiasis transmission [8,9] underscore the challenge of spatial heterogeneity in MDA success which is likely multifactorial, with snail distribution, abundance and genetics predicted to play a large role [10–12].

Snail control remains critical for breaking the cycle of SCH transmission. The risk of reinfection remains if infected snails continue releasing parasites into water sources, even with regular MDA campaigns. Indeed, WHO recommends the implementation of targeted snail control where feasible [9], with resolution WHA70.16 on the Global Vector Control Response (GVCR) 2017–2030 [13] calling on Member States to develop or adapt national vector control strategies and operational plans to align with the GVCR strategy. Implementation of effective snail control requires among other things an understanding of the occurrence of the responsible snails and ecological factors sustaining their distribution. However, the delineation of when and where transmission actually occurs is made difficult due to the longevity of schistosome infections in the human host coupled with the high vagility of humans [14], focal nature of infections and diverse species of reservoir hosts. Delimiting the species of snail present is important in informing not only the transmission dynamics (where human-infective cercariae are actually being produced and thus likely sites of active transmission), but also in the design of effective snail control interventions. For instance, surface molluscicides may not be effective against *Biomphalaria choanomphala*, a deep-water snail [15]. In addition, it helps in the timely identification/increased vigilance of invasive snail species that have the potential to influence transmission dynamics and/or exert negative economic and environmental impacts – as is the case of *Orientogalba viridis* in Malawi [16] and *Pomacea canaliculata* in Kenya [17].

Three main endemic regions for schistosomiasis are recognized in Kenya; the Coastal region (where the main spp. is *S. haematobium*), parts of Central and Lower Eastern region (both *S. haematobium* and *S. mansoni*) and the Lake Victoria basin area (where the main spp. is *S. mansoni*) (3). The distribution of the disease is determined, to a large extent, by the presence or absence of *Biomphalaria* and *Bulinus* snails, which act as the obligatory intermediate hosts for *S. mansoni* and *S. haematobium*, respectively. Several species of the shallow-water *Biomphalaria* snails are involved in transmission across sub-Saharan Africa [15]. Three species of *Biomphalaria* are involved in transmission in Kenya: 1) *Biomphalaria pfeifferi*, whose distribution includes tributaries feeding Lake Victoria, and in small impoundments and both seasonal and perennial streams throughout the country, except in the tropical lowland belt along the coast; 2) *Biomphalaria sudanica*, mainly found along the shores of Lake Victoria and Lake Jipe and their surrounding swamps; and 3) *Biomphalaria choanomphala*, a deeper water inhabitant of Lake Victoria [15,18]. In western Kenya, most malacological surveys have been confined around Lake Victoria [14, 19- 20] with a dearth of surveys in counties of western Kenya that do not border the lake, including Kakamega and Bungoma.

The prevalence of intestinal schistosomiasis in Kenya is highest (>50%) in the Mwea Irrigation Scheme in central Kenya and in the Lake Victoria basin in western Kenya [3]. The four counties of western Kenya (Kakamega, Bungoma, Vihiga and Trans Nzoia) have historically been considered non-endemic for schistosomiasis or with limited transmission [21,22], due to among other factors inadequate data on infections in humans. However, results of the recent (September 2021) granular mapping survey confirmed the presence of schistosomiasis and revealed that a total of 19 Wards in Kakamega and Bungoma counties had prevalence of schistosomiasis ≥10% [23]. Whereas the survey provided evidence that schistosomiasis was endemic in the 4 counties, it also revealed several foci with low number of infections raising the question as to whether transmission was occurring autochthonously or whether they were imported cases. To address the aforementioned gap and as part of additional surveillance activities, the presence, species and prevalence of *Schistosoma* spp. in snail vectors for schistosomiasis was assessed in 19 Wards with highest *S. mansoni* prevalence (>10%)

PLOS Neglected Tropical Diseases

in Kakamega and Bungoma counties of western Kenya. We hypothesized that (i) snail vectors of *Schistosoma* spp. are present in in the study areas and, (ii) that the prevalence of *Schistosoma* infection among host snails is greater than zero, providing evidence of autochthonous transmission of schistosomiasis in the study areas. The survey was part of interventions being piloted towards the interruption of transmission for schistosomiasis through END Fund's DIF M&E framework that envisages sustainable and context-appropriate snail control measures.

## Materials and methods

### Ethics statement

The study received approval from the Institutional Scientific Ethics Review Committee (ISERC) of the University of Eastern Africa Baraton (UEAB), approval # UEAB/ISERC/06/06/2023.

The authors confirm that the ethical policies of the journal, as noted on the journal's author guidelines page, have been adhered to.

### Study area and population

This study was conducted in Kakamega and Bungoma counties of western Kenya in February 2025 (Fig 1). Kakamega county is located at 00°20′N 34°46′E in the Western part of Kenya and covers an area of 3,051.3 Km$^2$. The county is served by two main rivers, Yala and Isiukhu, with several streams and springs. Bungoma county lies between latitude $000$ 28' and latitude $1^0$ 30' North of the Equator, and longitude $34^0$ 20' East and $35^0$ 15' East of the Greenwich Meridian and covers an area of 3,032.4 Km$^2$. The major physical features in Bungoma include Mt. Elgon, several hills (Chetambe, Sang'alo and Kabuchai), rivers (Nzoia, Kuywa, Sosio, Kibisi and Sio-Malaba/Malakisi), waterfalls such as Nabuyole and Teremi, and several dams. The temperatures in the 2 counties range from 18$^0$C to 29$^0$C. The region experiences two rainy

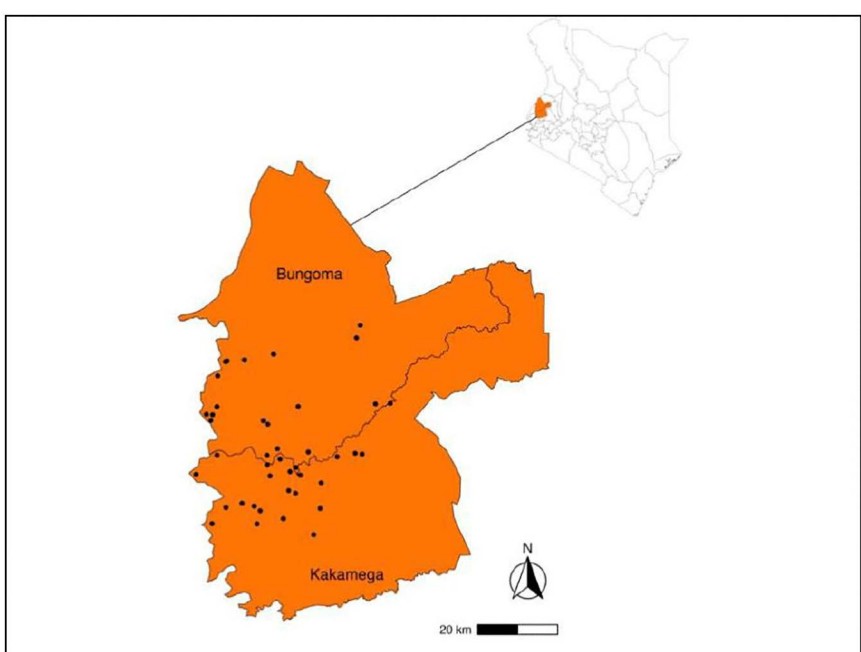

**Fig 1. Map of the geographic distribution of the 42 primary school regions that were sampled for snails in Kakamega and Bungoma counties in western Kenya.** https://www.gadm.org/download_country.html..

seasons, the long rains in March to July and short rains in August to October. The average annual rainfall is between 1,280–2,214 mm for Kakamega and 400–1,800mm for Bungoma.

The 2 counties have a combined population of approximately 3,538,149 based on the 2019 census [24]. The main ethnic communities are Luhya, Teso and Sabot. Communities in the region practice a mixture of both subsistence and cash crop farming, with sugar cane being the preferred large-scale cash crop. Other economic activities include cottage Industries, tea farming, horticulture, livestock farming, wholesale and retail trade, quarrying and mining. Based on results of the granular mapping [23], *S. mansoni* was the main schistosome species in the two counties. The very few cases of *S. haematobium* were confined to few pockets.

## Study sites sampled

The cross-sectional study involved 284 snail sampling sites selected from 42 primary school regions purposively selected in 19 Wards with the highest prevalence of *S. mansoni* in school-age children (SAC) in Kakamega and Bungoma counties (Fig 1) based on granular mapping data of 2021 [23]. Ten Wards with 22 primary school regions were included in Kakamega and 9 Wards with 20 primary school regions for Bungoma. Briefly, 2 primary school regions with the highest *S. mansoni* prevalence were selected in each Ward, 2 primary school regions that tied in *S. mansoni* prevalence were included and lastly 2 primary school regions that had the highest *S. haematobium* prevalence in SAC were also included. In each primary school region, water bodies including canals, rivers, streams, ponds, swamps/marshes and dams (Fig 2) where there was evidence of human-water contact activities were selected for snail sampling with the help of local community health promoters (CHPs). A river was considered as generally larger, deeper, with more continuous flow than a stream, while a stream was a smaller tributary that fed into a river. Each of the 284 sites sampled for snails in the primary school regions was visited once during the dry season (February 2025).

## Snail sampling

Four teams each comprising of 5 personnel conducted the sampling, 2 teams in Kakamega (11 sites per team) and 2 teams in Bungoma (10 sites per team). Working closely with the local CHP in charge of a village, an effort was made to identify the main water bodies at each selected primary school region, where several ideal sampling points (range: 1–15) were explored at each water body. Snails were sampled by four experienced collectors using a scoop that is a dip-net, with a mesh size of about 2 mm, supported by a metal frame and mounted on a wooden handle 2 metres long.

A timed direct search method was employed, where at each sampling point, a total of 20 random scoops were made on every sampling occasion. The area/size per sampling point was approximately 5 m² whereas lengths of 10 metres along streams, rivers and canals were used. Sampling time was fixed at 30 minutes at each water body and was carried out between 0900 hours and 1400 hours. The same collectors scooped for snails throughout so as to achieve some level of standardized sampling effort. The maximum time spent at each sampled site was not more than 2 hours, and each team covered 2 sites per day.

At each site, snails were separated, counted and identified to species-level based on shell morphological characteristics using standard taxonomic identification keys [25,26]. Briefly, morphological characters, included the general shape of the shell, shape of the whorls, number of coils and shape of the aperture. In order to determine the geographic distribution of snails the global positioning system (GPS) coordinates of all sampled habitats was geotagged using Kobo Collect Application (Cambridge, MA, USA). Data was collected in the field electronically using designed forms in Kobo Collect Application on android phones. All *Biomphalaria* and *Bulinus* snails from each sampled site were kept in separate well-labelled wide mouthed plastic containers with some aquatic vegetation from each of the sampled water bodies in order to maintain good aeration. All other snails were released back into the water body after examination. All *Biomphalaria* and *Bulinus* snails were transported to the Ministry of Health's Vector-borne and Neglected Tropical Diseases Laboratory in Kakamega and MoH's county Referral Hospital laboratory in Bungoma where they were screened for cercariae.

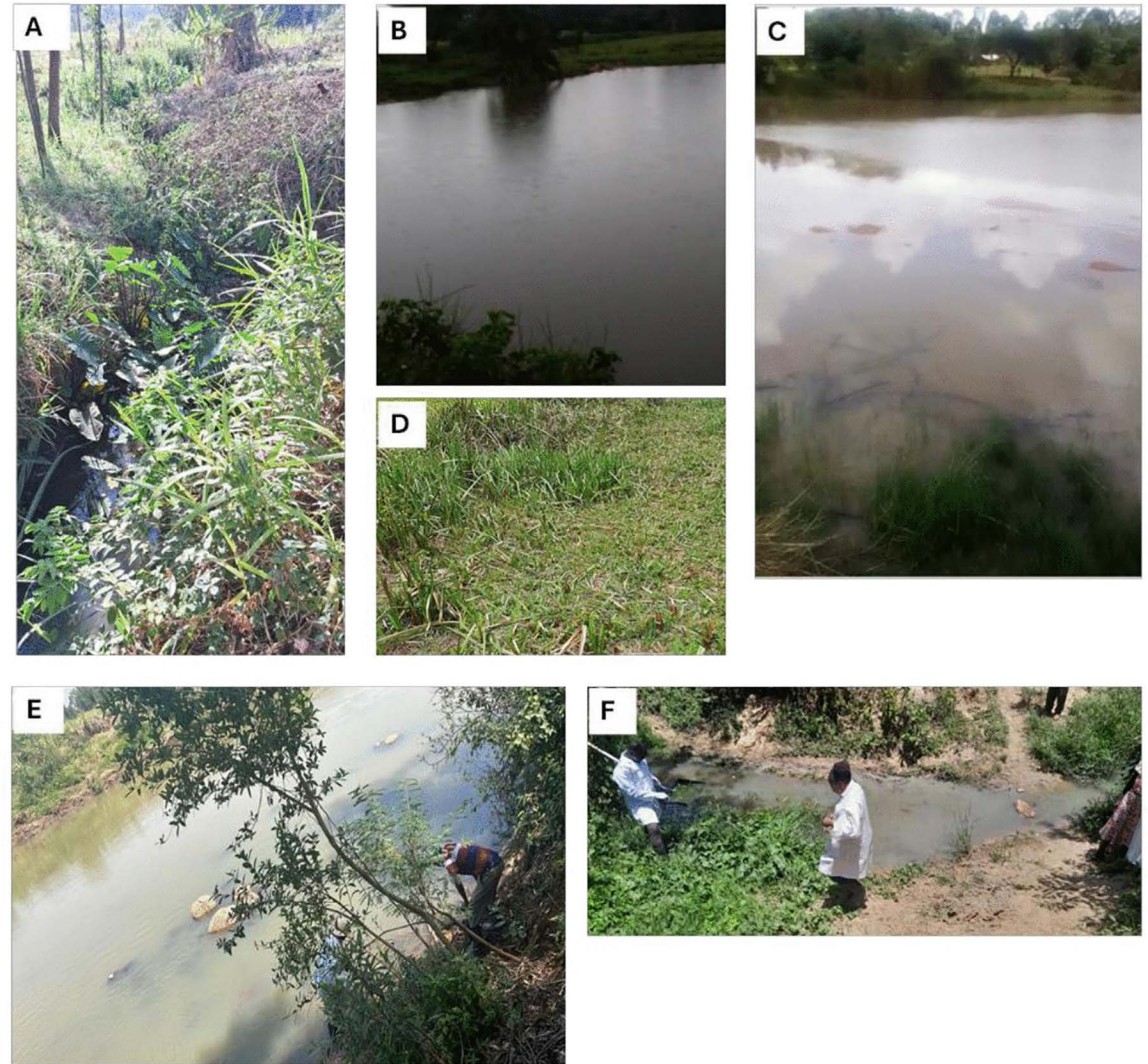

**Fig 2. Different types of water bodies from which snails were sampled in Kakamega and Bungoma: A) Canal, (B) Pond (Bukirimo in Kimaeti, Bungoma), (C) Dam (Kisawayi in Kabula, Bungoma, (D) Swamp/Marsh, (E) River (Nzoia in Kakamega), (F) Stream.**

## Snail screening for cercariae

To screen for patent infections, each snail was placed in an individual well of a 24-well plastic tissue culture plate containing 2 mL of bottled mineral water. The plate was exposed to indirect sunlight for 2 hours (1400–1600 hours) to induce cercarial shedding, a logistically driven timeframe that falls outside the optimal 0800–1300 hours emergence period for S. mansoni cercariae in Biomphalaria snails in this region [14]. This timeframe also overlaps with that for S. rodhaini, known to occur in this setting [14]. Based on the morphological criteria for identification of cercariae used in this study, the cercariae species could not be distinguished morphologically between S. mansoni and S. rodhaini or for any other Schistosoma sp.

Individual wells were then examined with the aid of a dissecting microscope for the presence or absence of cercariae. Upon staining with 1–2 drops of iodine, shed cercariae were identified to basic taxonomic groups using standard

identification keys [27]. A sub-set (~10 snails per site) of snails that shed mammalian cercariae and non-shedders were preserved in individual vials in absolute ethanol for molecular analysis later.

### Other variables

The presence or absence of human activity such as bathing, washing clothes, fishing and agricultural practices were recorded. The type of water body at each of the sampled sites was also recorded.

### Statistical analysis

Data was exported from the Kobo Collect server in xls format and statistical analysis performed using STATA v.18 (StataCorp.; College Station, TX, USA). P values < 0.05 were considered statistically significant. Most of our analysis focused on *B. pfeifferi* as opposed to *B. globosus* since the evidence (human infections) [23] strongly supported the role of *S. mansoni* in transmission in this setting, but we also include some analysis for *B. globosus* where relevant. Unless otherwise indicated, values are presented as means ± S.D. GPS coordinates for each primary school region were used to generate a map of the snail abundance and cercarial shedding sites using R statistical software (v4.4.0; R Core Team 2021). Data were checked for normality and homogeneity of variance. The mean *B. pfeifferi* snail abundance was computed as the total number of snails collected per primary school region divided by the number of snail sampling locations in the school region. A two-sample Mann-Whitney U-test (equivalent to Wilcoxon rank-sum test) was used to compare the difference in mean snail abundance and also prevalence of infection for *B. pfeifferi* between the sites in Kakamega and Bungoma counties. Prevalence of schistosome infection in snails (as determined through cercarial emergence) was computed as the proportion of snails shedding cercariae. To determine whether mean abundance of *B. pfeifferi* differed between primary school regions with human *S. mansoni* prevalence <10% and those with ≥10% (from granular mapping data), we performed generalized linear model (GLM) in STATA v18, with the random effect of school and using the family negative binomial and link as logit. To determine the likelihood of finding an infected snail and the total number of snails, we correlated the number of *B. pfeifferi* positive for *Schistosoma* sp. cercariae with the total number of *B. pfeifferi* snails at primary school region using spearman correlation ($r_s$). To determine if the abundance of *B. pfeifferi* snails at a particular primary school regions was associated with schistosome infection rates in school-age children (granular mapping data of September 2021), we correlated the mean number of snails collected at a school region with the initial prevalence in SAC at the corresponding region using spearman correlation ($r_s$). The Kruskal-Wallis test with post-hoc Dunn's test was used to compare the difference in mean *B. pfeifferi* snail abundance among the different types of water sources.

## Results

### Snail abundance

A total of 4,245 snails were collected across the 284 sites in the 42 primary school regions, with all but one school region yielding snails. A rich snail species diversity was observed in the sites sampled, with a total of 7 freshwater snail species putatively identified. Overall, *B. pfeifferi* was the dominant species (78.7%) collected followed by *Lymnaea natalensis* (15.6%). Other snail species identified included *Bulinus globosus*, *Melanoides tuberculata*, *Physa acuta*, *Pila ovata* and *Ceratophallus* spp. (Table 1 and S1 Data). No *Biomphalaria sudanica* snails were found at any of the sampled locations.

The mean abundance of *B. pfeifferi* was 1.8-fold higher in sampled sites in Kakamega (102 ± 131) compared to sites in Bungoma county (56 ± 103) (Z = -2.169, P = 0.0301). The converse was true for the mean abundance of *B. globosus* which was 1.5-fold higher in sites in Bungoma (3 ± 6) relative to sites in Kakamega county (2 ± 6) (Z = 2.334, P = 0.0196).

The mean *B. pfeifferi* abundance was lower in primary school regions with ≥10% *S. mansoni* prevalence compared to regions with <10% *S. mansoni* prevalence (Y = -1.54x + 3.91, P = 0.011), but was highly variable among regions. For instance, at Okanya primary school region in Bungoma, one of the regions with <10% *S. mansoni* prevalence it was

**Table 1. Different freshwater snail species identified across sampled sites within the 42 primary school regions in Kakamega and Bungoma counties in western Kenya.**

| Type of snail/species | Kakamega county | | Bungoma county | | | |
|---|---|---|---|---|---|---|
| | No. of snails by species | Relative abundance (%)$ | No. of snails by species | Relative abundance (%)$ | Overall Total No. of snails | Overall Relative abundance (%, 95% CI)β |
| *Biomphalaria pfeifferi* | 2235 | 82.4 | 1106 | 72.1 | 3341 | 78.7 (77.5-79.9) |
| *Bulinus globosus* | 29 | 1.1 | 59 | 3.8 | 88 | 2.1 (1.7-2.5) |
| *Lymnaea natalensis* | 377 | 13.9 | 287 | 18.7 | 664 | 15.6 (14.6-16.7) |
| *Melanoides tuberculata* | 31 | 1.1 | 79 | 5.1 | 110 | 2.6 (2.1-3.1) |
| *Physa acuta* | 39 | 1.4 | 0 | 0 | 39 | 0.9 (0.6-1.2) |
| *Pila Ovata* | 0 | 0 | 2 | 0.1 | 2 | 0.05 (0.006-0.17) |
| *Ceratophallus* | 0 | 0 | 1 | 0.1 | 1 | 0.02 (0.0006-0.1313) |
| Total | 2711 | | 1534 | | 4245 | |

$*Relative species abundance per county is derived from number of snails by species divided by the total number of snails collected in each county (n = 2711 for Kakamega; n = 1534 for Bungoma).*

β *Overall Relative abundance is derived as the number of snail species for the two counties divided by the total number of snails collected (n = 4245).*

high (mean abundance = 115 snails), while the mean abundance was low (1 snail) in Mahola primary school region in Kakamega, one of the regions with ≥10% *S. mansoni* prevalence.

A summary of the total number of snails, number of *B. pfeifferi* and *B. globosus* and the number of snails that shed any type of cercariae in Kakamega and Bungoma Counties by Ward is presented in Table 2.

A total of 1,106 and 2,235 *B. pfefferi* snails were collected in Bungoma and Kakamega counties, respectively (Table 2). Maraka Ward in Bungoma County had the highest number of snails that shed mammalian schistosome cercariae (Table 2). In Kakamega county, the highest number of snails that shed mammalian schistosome cercariae was 18 and 16 for Bunyala West Ward and Lusheya-Lubinu Wards, respectively (Table 2).

**Spatial distribution of snails and *Schistosoma* cercarial shedding**

*B. pfeifferi* were found in 36 out of the 42 primary school regions (85.7%) distributed across all the 19 Wards that were selected for the survey while *B. globosus* were recovered in only 8 primary school regions (19%) located in 6 Wards (Table 3). High *B. pfeifferi* abundance was noted in school regions closer to river Nzoia and associated tributaries (Fig 3). *B. pfeifferi* was collected at all the 8 school regions where *B. globosus* was present, except for 1 school region (Mikokwe in Bumula Ward). The highest number of *B. pfeifferi* (605 snails) was recorded at Indangalasia school region (Table 3) in Lusheya-Lubinu Ward, which interestingly also had both a high number of positive snails (16 snails) and high *S. mansoni* prevalence (35%) in SAC from granular mapping. The highest number of *B. globosus* (29 snails) was recorded at Lutasio school region in Khalaba Ward (Table 3), Kakamega county.

Overall, of the 3,341 *B. pfeifferi* snails collected, 87 (2.6%, 95% CI: 2.07-3.14) shed *Schistosoma* sp. cercariae. Cercarial shedding was recorded in 13 out of 42 school regions (31%; 95% CI: 17–45%) and shedding was highly variable among sites (Table 3). Out of these 13 school regions where shedding occurred, 7 were in Bungoma and 6 in Kakamega county (Fig 3). With respect to having high overall number of *B. pfeifferi* snails that shed cercariae, two school regions stood out from the rest, Kisembe in Kakamega county where 18 snails shed and Maraka PEFA in Bungoma county where 19 snails shed (Table 3). There was no difference in prevalence of *Schistosoma* sp. infection in *B. pfeifferi* between the school regions in Kakamega and Bungoma counties (Z = 1.037, P = 0.2999). 56 out of 2,235 (2.5%) and 31 out of 1,106 (2.8%) *B. pfeifferi* snails collected from Kakamega and Bungoma counties, respectively, shed *S. mansoni* cercariae.

**Table 2. Total number of snails, number of *Biomphalaria pfeifferi* and *Bulinus globosus* and the number of snails that shed any type of cercariae in Kakamega and Bungoma Counties by Ward.**

| County | Ward | No. of sampling points/ sites | Total snails (regardless of species) | *B. pfeifferi* | No. of *B. pfeifferi* that shed cercariae | *B. globosus* | No. of *B. globosus* that shed cercariae | No. of snails that shed mammalian schistosome cercariae |
|---|---|---|---|---|---|---|---|---|
| Bungoma | Bumula | 20 | 234 | 137 | 0 | 31 | 0 | 0 |
| Bungoma | East Sang'alo | 18 | 151 | 134 | 0 | 0 | 0 | 0 |
| Bungoma | Kabula | 11 | 102 | 19 | 6 | 15 | 0 | 3 |
| Bungoma | Kimaeti | 13 | 42 | 27 | 0 | 0 | 0 | 0 |
| Bungoma | Kimilili | 19 | 45 | 14 | 5 | 4 | 4 | 5 |
| Bungoma | Maraka | 3 | 50 | 40 | 21 | 0 | 0 | 21 |
| Bungoma | Marakaru/Tuuti | 19 | 217 | 124 | 4 | 2 | 0 | 0 |
| Bungoma | Musikoma | 7 | 551 | 525 | 0 | 0 | 0 | 0 |
| Bungoma | Siboti | 27 | 142 | 86 | 5 | 7 | 0 | 1 |
| Kakamega | Bunyala East | 11 | 286 | 277 | 0 | 0 | 0 | 0 |
| Kakamega | Bunyala West | 19 | 189 | 160 | 18 | 0 | 0 | 18 |
| Kakamega | East Wanga | 8 | 70 | 70 | 0 | 0 | 0 | 0 |
| Kakamega | Khalaba | 8 | 224 | 176 | 7 | 29 | 0 | 7 |
| Kakamega | Kholera | 20 | 119 | 60 | 11 | 0 | 0 | 7 |
| Kakamega | Lusheya-Lubinu | 20 | 675 | 614 | 18 | 0 | 0 | 16 |
| Kakamega | Malaha-Isongo-Makunga | 22 | 405 | 376 | 11 | 0 | 0 | 7 |
| Kakamega | Mayoni | 22 | 354 | 233 | 0 | 0 | 0 | 0 |
| Kakamega | Mumias North | 11 | 134 | 110 | 1 | 0 | 0 | 1 |
| Kakamega | Namamali | 6 | 255 | 159 | 4 | 0 | 0 | 0 |
| Total | | 284 | 4245 | 3341 | 111 | 88 | 4 | 87 |

There was no correlation between the number of *B. pfeifferi* positive for *Schistosoma* sp. cercariae and the total number of *B. pfeifferi* snails ($r_s = 0.0864$, P = 0.5853). Similarly, neither the total number nor the mean number of *B. pfeifferi* collected at the school regions was significantly correlated with *S. mansoni* prevalence in SAC, ($r_s = 0.1601$, P = 0.3099) and ($r_s = 0.1490$; P = 0.3447), respectively (Fig 4).

Prevalence of *Schistosoma* sp. cercariae in *B. pfeifferi* varied spatially and ranged from 1.2-72.0% among school regions where infected snails were found (Table 3). Prevalence of *Schistosoma* sp. cercariae in *B. pfeifferi* was highest at the Kisembe primary school region (72%) in Bunyala West Ward, Kakamega county, which also recorded a high prevalence of *S. mansoni* (40.7%) among SAC in the granular mapping survey [23]. Other school regions that had high cercarial shedding included Maraka PEFA primary (61.3%) in Maraka Ward and Kambini primary (38.5%) in Kimilili Ward, both in Bungoma county (Table 3). The prevalence of *S. mansoni* among SAC in the granular mapping survey was 50% and 28.3% for Maraka PEFA and Kambini schools, respectively. Nevertheless, the heterogeneity between school regions with high *S. mansoni* prevalence in SAC in relation to cercarial shedding in *B. pfeifferi* snails is worth noting as the pattern is difficult to succinctly characterize. For instance, while Kamuli primary school region recorded *S. mansoni* prevalence of 61.7% among SAC with a total of 135 *B. pfeifferi* collected, none of the snails shed cercariae. On the other hand, Kikwechi primary school region which recorded *S. mansoni* prevalence of 44.6% among SAC with a total of 83 *B. pfeifferi* collected, only 1 snail shed cercariae (Table 3). No *B. pfeifferi* snails were collected around Ebwaliro primary school region which interestingly recorded *S. mansoni* prevalence of 45% among SAC in the granular mapping survey.

**Table 3. The total number of *Biomphalaria pfeifferi* and *Bulinus globosus* collected, and the number of *B. pfeifferi* positive for *Schistosoma* sp. cercariae from sites within the 42 primary school regions sampled.**

| School region name | County | Latitude | Longitude | *S. mansoni* Prevalence (%) in SAC (2021 survey) | No. of Sampled points/ sites | No. of *B. pfeifferi* (Mean abundance)$ | No. of *B. pfeifferi* (+) for *Schistosoma* sp. cercariae, *n* (%) | 95% CI for prev | *B. pfeifferi* that shed other digenetic cercariae | No. of *B. globosus* |
|---|---|---|---|---|---|---|---|---|---|---|
| Bukirimo | Bungoma | 0.55533 | 34.3839 | 20.0 | 5 | 0 (0) | 0 (0) | NA | 0 | 0 |
| Chemche ACK | Bungoma | 0.57298 | 34.5914 | 3.3 | 14 | 130 (10) | 0 (0) | NA | 0 | 0 |
| Fuchani | Bungoma | 0.47054 | 34.6143 | 18.3 | 2 | 4 (2) | 0 (0) | NA | 0 | 0 |
| Kakichuma R.c | Bungoma | 0.69061 | 34.5356 | 18.5 | 10 | 41 (5) | 0 (0) | NA | 0 | 2 |
| Kambini | Bungoma | 0.72714 | 34.7227 | 28.3 | 9 | 13 (2) | 5 (38.5) | 12.02-64.9 | 0 | 0 |
| Kikwechi | Bungoma | 0.67515 | 34.4307 | 44.6 | 9 | 83 (10) | 1 (1.2) | -1.1-3.6 | 3 | 0 |
| Kitabisi R.C | Bungoma | 0.64187 | 34.4091 | 16.7 | 8 | 27 (4) | 0 (0) | NA | 0 | 0 |
| Maraka PEFA | Bungoma | 0.57999 | 34.7989 | 50.0 | 2 | 31 (16) | 19 (61.3) | 44.1-78.4 | 0 | 0 |
| Matili F.Y.M | Bungoma | 0.75563 | 34.7314 | 30.0 | 10 | 1 (1) | 0 (0) | NA | 0 | 4 |
| Mikokwe ACK | Bungoma | 0.54166 | 34.3939 | 33.3 | 10 | 0 (0) | 0 (0) | NA | 0 | 9 |
| Mukhuma | Bungoma | 0.46313 | 34.5208 | 25.4 | 6 | 11 (2) | 1 (9.1) | -7.9-26.1 | 0 | 0 |
| Mwikhupo | Bungoma | 0.43584 | 34.5863 | 16.7 | 2 | 0 (0) | 0 (0) | NA | 0 | 0 |
| Mwiyenga | Bungoma | 0.57265 | 34.4075 | 16.7 | 2 | 0 (0) | 0 (0) | NA | 0 | 0 |
| Netima | Bungoma | 0.67754 | 34.4699 | 31.7 | 14 | 73 (6) | 0 (0) | NA | 1 | 2 |
| Nzoia PEFA | Bungoma | 0.57871 | 34.7656 | 30.0 | 1 | 9 (9) | 2 (22.2) | 4.9-49.4 | 0 | 0 |
| Okanya | Bungoma | 0.54079 | 34.5127 | 9.8 | 4 | 457 (115) | 0 (0) | NA | 0 | 0 |
| Sango R.C | Bungoma | 0.67371 | 34.4273 | 43.3 | 13 | 13 (1) | 1 (7.7) | -6.8-22.2 | 3 | 5 |
| Wamunyiri | Bungoma | 0.47807 | 34.5442 | 21.7 | 3 | 8 (3) | 2 (25.0) | -5-55 | 3 | 15 |
| Wekelekha | Bungoma | 0.53298 | 34.5227 | 20.0 | 3 | 68 (23) | 0 (0) | NA | 0 | 0 |
| Wesimikha | Bungoma | 0.55424 | 34.398 | 43.3 | 10 | 137 (14) | 0 (0) | NA | 0 | 22 |
| Chekata | Kakamega | 0.46769 | 34.7199 | 10.0 | 3 | 0 (0) | 0 (0) | NA | 0 | 0 |
| Ebubambula | Kakamega | 0.3455 | 34.4283 | 20.0 | 10 | 170 (17) | 0 (0) | NA | 0 | 0 |
| Ebwaliro | Kakamega | 0.3086 | 34.498 | 45.0 | 3 | 0 (0) | 0 (0) | NA | 0 | 0 |
| Eshibanze Muslim | Kakamega | 0.41697 | 34.5277 | 30.0 | 13 | 7 (1) | 0 (0) | NA | 0 | 0 |
| Eshisenye | Kakamega | 0.34334 | 34.6411 | 23.3 | 13 | 265 (21) | 7 (2.6) | 0.7-4.6 | 0 | 0 |
| Ichinga | Kakamega | 0.33807 | 34.5054 | 32.2 | 5 | 60 (12) | 0 (0) | NA | 0 | 0 |
| Indangalasia | Kakamega | 0.30914 | 34.3967 | 35.0 | 13 | 605 (47) | 16 (2.6) | 1.4-3.9 | 1 | 0 |
| Isango | Kakamega | 0.28438 | 34.6263 | 26.7 | 9 | 111 (13) | 0 (0) | NA | 4 | 0 |
| Kamashia | Kakamega | 0.32021 | 34.5575 | 35.0 | 4 | 9 (3) | 0 (0) | NA | 1 | 0 |
| Kamuli | Kakamega | 0.41838 | 34.5964 | 61.7 | 4 | 135 (34) | 0 (0) | NA | 0 | 0 |
| Khaunga | Kakamega | 0.37724 | 34.5854 | 25.0 | 4 | 39 (10) | 0 (0) | NA | 0 | 0 |
| Kisembe | Kakamega | 0.40112 | 34.6433 | 40.7 | 15 | 25 (2) | 18 (72.0) | 54.4-89.6 | 0 | 0 |
| Lutasio | Kakamega | 0.45445 | 34.5501 | 35.0 | 5 | 86 (18) | 7 (8.1) | 2.3-13.9 | 0 | 29 |

*(Continued)*

**Table 3.** (Continued)

| School region name | County | Latitude | Longi-tude | *S. mansoni* Prevalence (%) in SAC (2021 survey) | No. of Sampled points/ sites | No. of *B. pfeifferi* (Mean abundance)$ | No. of *B. pfeifferi* (+) for *Schistosoma* sp. cercariae, *n* (%) | 95% CI for prev | *B. pfeifferi* that shed other digenetic cercariae | No. of *B. globosus* |
|---|---|---|---|---|---|---|---|---|---|---|
| Mahola | Kakamega | 0.38343 | 34.5696 | 18.3 | 4 | 31 (8) | 0 (0) | NA | 0 | 0 |
| Makhima | Kakamega | 0.46579 | 34.7352 | 0.0 | 3 | 78 (26) | 0 (0) | NA | 0 | 0 |
| Makokhwe | Kakamega | 0.42643 | 34.5726 | 35.0 | 3 | 90 (30) | 0 (0) | NA | 0 | 0 |
| Munanga | Kakamega | 0.3549 | 34.4646 | 40.0 | 12 | 63 (6) | 1 (1.6) | -1.5-4.7 | 0 | 0 |
| Mungakha | Kakamega | 0.46342 | 34.4077 | 45.0 | 3 | 137 (46) | 0 (0) | NA | 4 | 0 |
| Namayiakalo | Kakamega | 0.42008 | 34.3604 | 25.0 | 3 | 22 (8) | 0 (0) | NA | 0 | 0 |
| Sheikh Khalifa | Kakamega | 0.34861 | 34.492 | 25.0 | 6 | 50 (9) | 0 (0) | NA | 0 | 0 |
| Shiyabo | Kakamega | 0.44111 | 34.5214 | 26.2 | 7 | 53 (8) | 7 (13.2) | 4.1-22.3 | 4 | 0 |
| Weremba | Kakamega | 0.45983 | 34.6792 | 31.7 | 5 | 199 (40) | 0 (0) | NA | 0 | 0 |

$ Mean abundance of *B. pfeifferi* derived by dividing total number of snails by the number of sampled points/sites, and value rounded to the nearest whole snail.

Of the 88 *B. globosus* collected, none shed *Schistosoma* sp. cercariae cercariae.

## Infection of *Biomphalaria pfeifferi* and *Bulinus globosus* with other digenetic trematodes

A total of 24 *B. pfeifferi* snails shed non-*Schistosoma* cercariae, which included Xiphidio cercariae, Echinostome cercariae and Amphistome cercariae.

Four out of the 88 *B. globosus* snails collected at Matili F.Y.M school region in Kimilili Ward, Bungoma county shed echinostome cercariae. None of the snails that shed cercariae were found to have double infections (e.g., to be simultaneously shedding two different kinds of cercariae).

A summary of the different types of cercariae shed by *B. pfeifferi* and *B. globosus* is presented in Table 4**.**

## Distribution of snails of *Biomphalaria pfeifferi* snails among water bodies sampled in Kakamega and Bungoma counties

The total and relative *B. pfeifferi* abundance was highest in streams relative to the other types of water bodies sampled (Fig 5).

The mean *B. pfeifferi* snail abundance was significantly different across the different types of water bodies sampled (H = 25.5, P = 0.0001). While the mean abundance of *B. pfeifferi* was similar between streams and swamps/marshes, the proportion by type of water body was highest in streams (80.3%) (Table 5). The mean *B. pfeifferi* abundance at the streams was higher compared to rivers (P = 0.005) and ponds (P = 0.005) (Table 5). Dams/water pans did not yield any *B. pfeifferi* snails.

## Human-water contact activities

Several human-water contact activities were observed and noted among the different water bodies sampled. The activities included swimming, bathing, and fetching water for drinking and laundry. Other observed activities were irrigation, fishing and sand harvesting.

PLOS Neglected Tropical Diseases

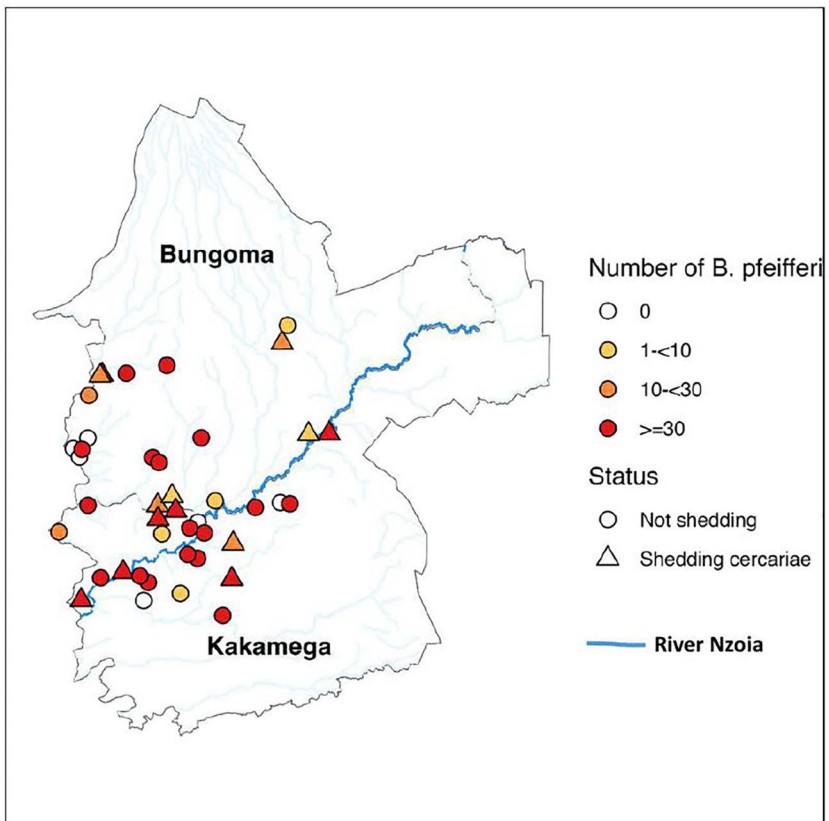

**Fig 3. Map showing the geographic distribution of the 42 primary school regions and abundance of *Biomphalaria* snails within the school regions in Kakamega and Bungoma counties. https://www.gadm.org/download_country.html.**

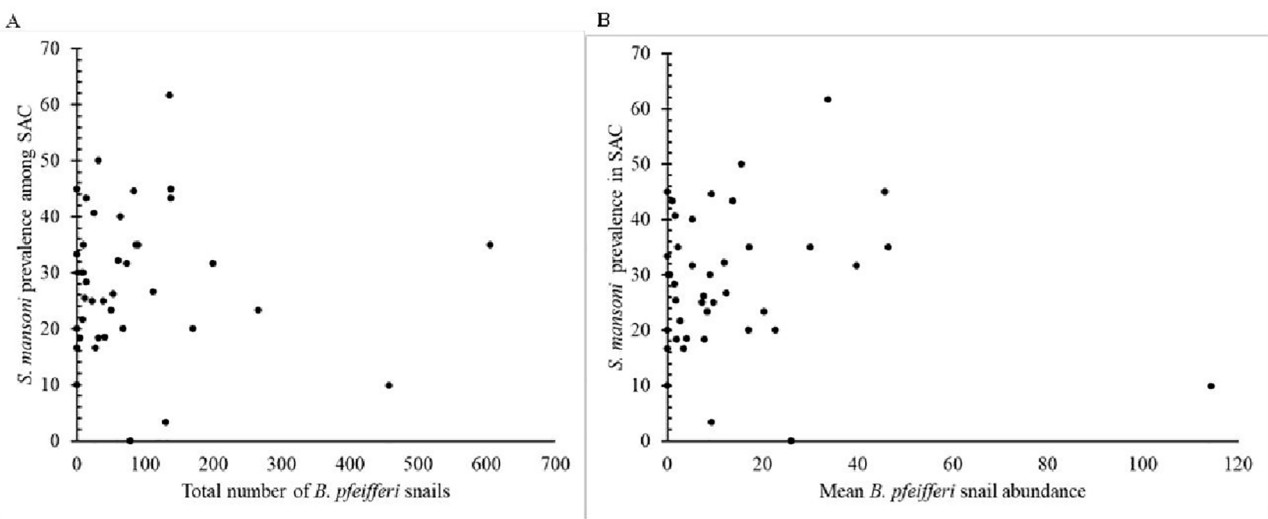

**Fig 4. Correlation between Total number (A) and mean number (B) of *Biomphalaria pfeifferi* snails and prevalence of *Schistosoma mansoni* among school-age children (SAC).**

**Table 4. Different types of cercariae shed by *Biomphalaria pfeifferi* and *Bulinus globosus*.**

| Snail species | Total number | No. of snails shedding cercariae | | | |
|---|---|---|---|---|---|
| | | Schistosome cercariae | Xiphidio | Echinostome | Amphistome |
| *B. pfeifferi* | 3341 | 87 | 16 | 7 | 1 |
| *B. globosus* | 88 | 0 | 4 | 0 | 0 |

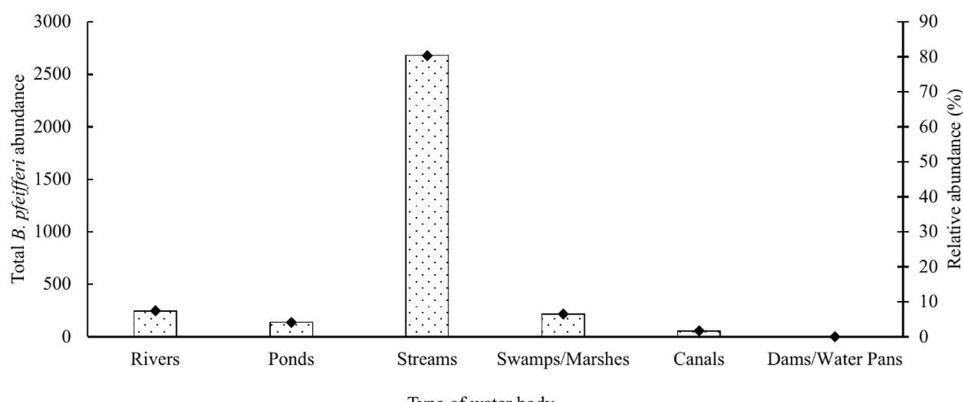

**Fig 5. Total abundance of *B. pfeifferi* snails in the different types of water sources in Kakamega and Bungoma county, western Kenya.**

**Table 5. Distribution of *Biomphalaria pfeifferi* snails among water bodies in Kakamega and Bungoma counties.**

| Type of water body | No. of points on water body sampled | Total snail abundance | Proportion (%) of snails by water body | Mean snail abundance[1] | Total No. of snails shedding *Schistosoma* sp. cercariae in each water body type |
|---|---|---|---|---|---|
| Rivers | 64 | 246 | 7.4 | $4 \pm 10^b$ | 37 |
| Ponds | 25 | 138 | 4.1 | $6 \pm 15^b$ | 3 |
| Streams | 170 | 2683 | 80.3 | $16 \pm 35^a$ | 38 |
| Swamps/Marshes | 11 | 218 | 6.5 | $20 \pm 31^a$ | 9 |
| Canals | 1 | 56 | 1.7 | $56^a$ | 0 |
| Dams/Water Pans | 13 | 0 | 0 | 0 | 0 |

[1] Different lowercase letters indicate significant difference for type of water body ($P < 0.05$). Values are means ± SD (rounded off to whole numbers)

## Discussion

A prerequisite to implementation of effective vector control strategies requires among other things an understanding of the vectors and ecological factors sustaining their distribution. The present study confirmed the presence, species and infectionj of *Biomphalaria* and *Bulinus* snails with *Schistosoma* sp. answering the lingering and intriguing question of autochthonous transmission of intestinal schistosomiasis in Kakamega and Bungoma counties. Few malacology studies have been conducted in this setting. Whereas one study documented high snail species composition and diversity in the bordering county of Trans Nzoia and suggested their potential as vectors for trematodes [28], this study looked at land and not freshwater snails. Both *B. pfeifferi* and *B. sudanica* have been reported in the neighboring county of Busia [29],

while *Biomphalaria* and *Bulinus* snails were found in Uganda's Mt. Elgon area [30]. Several *Biomphalaria* spp. including *B. sudanica, B. choanomphala, B. smithi, B. stanleyi, B. angulosa* and *B. pfeifferi* transmit *S. mansoni* in East Africa [31], but *B. pfeifferi* is arguably the most important due to its widespread distribution and its relatively high susceptibility to schistosome infections [32]. *B. pfeifferi* is known to prefer shallow/swampy water, with plant detritus as a substratum [33]. We found *B. pfeifferi* at 36 out of the 42 primary school regions (85.7%), with a preponderance for occurrence in streams and rivers. River Nzoia and its tributaries are the main sources of infection. There was no obvious indication that the closely related species *B. sudanica* plays a role in *S. mansoni* transmission in any of the sites sampled as no snails of this species were found. The distribution of snails across sampled locations was heterogeneous, and this may be explained by a combination of environmental and ecological factors (water physicochemical conditions, water speed, pollution, habitat type) [34], seasonal factors (dry season, heavy rains) [35], biological interactions (predators and competition) [36,37] and human-related factors (environmental modifications, pollution etc) [38,39] known to affect snail distribution.

Four types of digenetic trematode cercariae (*Schistosoma* sp., xiphidio, echinostome and amphistome) were shed by *B. pfeifferi* snails in our study, reflecting a fairly narrow biodiversity. Although antagonistic trematode interactions in snail hosts can affect schistosomiasis transmission [40], this likely doesn't influence transmission dynamics in this setting given the low trematode biodiversity and absence of co-infections in *B. pfeifferi*. A low proportion of *B. pfeifferi* (2.6%) in our study shed *Schistosoma* sp. cercariae, likely to be those of *S. mansoni*, although we cannot rule out cercariae of *S. rodhaini*, known to occur in this setting [14]. *S. mansoni* cercariae emerge diurnally, peaking at dawn and midday [14] when their putative hosts enter the water and are available for infection. *S. mansoni* and *S. rodhaini* simultaneously infect *Biomphalaria* snails [14,41], with *S. rodhaini*'s entire shedding period extending from 1430–1030 hours [41], which overlaps with the shedding period in our study. However, based on the associated epidemiology (confirmed *S. mansoni* infections among SAC [23]) and, to some extent, the shedding pattern it is most likely that these were *S. mansoni* rather than *S. rodhaini*. The low proportion of cercarial shedding in our study is consistent with other studies that reported shedding ranging from below 1% to about 3% [14,19,42,43,44–46], although higher prevalence of 13.1% [47] and 24.4% [48] have also been reported. Absence of shedding even in areas of high transmission is also documented [49,50]. Nevertheless, it should be noted that even minimal exposure to cercariae-infested water can result in infection [51], even with only a few snails shedding [52]. Few infected snails shedding cercariae daily can potentially translate into a large number of human infections.

Several explanations may be advanced for the low numbers of snails shedding *Schistosoma* cercariae in our study. First, our study sampled snails in the afternoon (1400–1600 hrs), which may have missed the peak shedding time. Relative to other studied populations such as *S. mansoni* from Cameroon that typically undergo peak emergence around noon or later [53], peak cercarial emergence of *S. mansoni* in *Biomphalaria* snails from Lake Victoria in western Kenya occurs earlier between 0800–1300 hours [14]. Other explanations proposed for low shedding and which were discussed in detail in our previous work [19] include: (i) low infection rates or brief time points (pulses) for release of cercariae [54,55]; (ii) sampling challenges - schistosomiasis is highly localized, and the wide distribution of snails complicates identifying hotspots of infection; (iii) environmental and biological factors - infection may be suppressed by environmental contaminants or organisms like rotifers [56]; (iv) pre-patent infections [57,58] that may be underestimated using the cercarial emergence technique [59]; and (v) seasonal influence - snail infection and cercarial output vary by season [44,49], with highest cercarial output expected in the drier and hotter periods [60,61] when human-water contact is longer and more frequent and with snail concentration in the fewer water sources relative to other seasons [62]. Given that the present study was conducted in February, which is drier and hotter, the 2.6% cercarial output which is higher relative to the prevalence reported in several other studies may be reflective of this higher cercarial transmission potential in drier periods. Our findings corroborate the pattern observed for *Bulinus* spp. in Iringa, Tanzania, where *Schistosoma* spp. cercarial emergence was higher during the dry season [61], but contrasts with other studies from Tanzania [35,63] and South Africa [45] that reported higher cercarial emergence during the rainy season (presumably when temperatures are lower). Elsewhere,

Gouvras and others [64] observed that shedding by *Biomphalaria* did not show any variation between dry and rainy seasons. Seasonal variations in cercarial shedding have implications for the design of control interventions. For instance, a mollusciciding programme should ideally be planned in a way that maximizes decrease in transmission at a minimum cost [65]. Therefore, proper timing of mollusciciding is important as it would not be advisable to apply molluscicides when there is very little or no transmission occurring. The WHO recommends that MDA should be conducted when the risk of reinfection is low [66], because higher reinfection rates cause the prevalence of infection to return to pre-treatment levels. To complement this, it is advised to reduce snail populations by applying molluscicides at least 5–7 weeks before MDA - a strategy that ensures reduced snail population and subsequently, a lower risk of reinfection [66].

In terms of transmission potential, a larger snail population means the probability of a parasite encountering and infecting a snail also increases, leading to a greater likelihood of finding infected snails [42]. Interestingly and rather counter-intuitive, we did not find an association between the number of *B. pfeifferi* positive for *S. mansoni* and the total number of *B. pfeifferi* snails. Other studies on *S. haematobium* in *Bulinus truncatus* along the Nile delta [60,67] reported similar findings. However, Gouvras and others [64] found that the number of shedding snails was positively associated with abundance of *B. sudanica.* Our finding suggests that while snail abundance is important for understanding disease transmission, infection prevalence depends not just on snail abundance but also on parasite density in water, snail susceptibility to different *Schistosoma* spp. and environmental conditions [20,68,69]. Furthermore, our study did not find an association between the prevalence of *S. mansoni* infection in *B. pfeifferi* and prevalence in humans (SAC), consistent with the study by Mutuku and others [42], but in contrast to Wang *et al.* [70]. The disparity in infection levels between snails and humans is supported by modeling studies that show that only a small percentage of snails need to be infected and shedding for high levels of human infection to occur [71]. We cannot rule out the fact that the disparity in infection levels between snails and humans in our study could simply be due to temporal variation, given that prevalence data in SAC was collected back in September 2021.

It is noteworthy mentioning that implementation of snail control activities requires competencies in malacology. This is especially important considering the limited capacity in snail control that exists in endemic countries, contributed in part by the priority given to preventive chemotherapy and the abandonment of snail control. Leveraging new tools and technologies such as artificial intelligence (AI) will go a long way in building capacity for the surveillance and evaluation of snail control activities, especially where interruption of transmission is envisaged. In this regard, the WHO App. for the identification of intermediate host snails [72] currently under development will be an invaluable resource in predicting snail species especially where there is uncertainty and has great potential to enhance snail control interventions in poor resource settings. In addition, to facilitate implementation of snail control, WHO published on specific standardized procedures and criteria for efficacy testing and evaluation, and a manual on the use of molluscicides in the field [66], and plans to strengthen capacities of health staff in medical malacology to reinforce snail control activities in countries [73].

The current study was not without limitations. First, we did not keep and maintain non-shedding snails for screening at a later time point for detection of snails that may have harbored pre-patent infections at the time of initial collection. It is plausible that some of the non-shedders may have harbored pre-patent infections and therefore the reported prevalence of infection in snails may have been underestimated. Pre-patent infections can substantially exceed patent infections [58] and are known to last for several weeks with only a proportion of snails reaching the stage of cercarial shedding [57]. However, methods such as crushing snails or the maintenance of non-shedding snails in search of larvae to clarify such pre-patent infections are unsuitable for large-scale monitoring and were beyond the scope of the current survey. Third, identification of cercariae was based on morphological characteristics which allow for identification of cercariae to the *Schistosoma* genus level, and species was not confirmed by molecular techniques. The suggestion for *S. mansoni* cercariae though not confirmed, is corroborated by the observed infections among SAC in the study areas [23]. Fourth, data on the abundance and infection of *Schistosoma* sp. *mansoni* in snails and that of *S. mansoni* infection in SAC was not contemporaneous, which may have weakened chances of detecting any useful associations. Fifth, we did not employ dredging to look for deep-dwelling snails, especially in dams.

Our findings of snail vectors and their shedding of *S. mansoni* cercariae provide solid evidence for autochthonous transmission in Kakamega and Bungoma counties of western Kenya. This argues for incorporation of focal snail control where feasible to complement chemotherapy alongside other interventions to accelerate interruption of transmission in these areas. The multiple water contact sites add to the complexity of vector control required in this setting. Whereas the impact of chemical-based molluscicides on schistosomiasis transmission has been documented, this approach is fraught with challenges among them cost, toxicity and the need for regular application [74]. Novel solutions that offer a lower cost, more sustainable option, for instance, simple measures such as fencing or bridging to keep people, particularly children, away from the water can be implemented in the confirmed transmission sites. These can be complemented by the provision of alternative facilities for water contact activities, such as swimming and laundry, where possible. Other cost-effective measures for safe water supply, including collection of rainwater can also be employed to minimize contact with transmission sites. Leveraging initiatives such as citizen science projects [75] could support consistent monitoring of snail species abundance and distribution as well as increasing awareness and participation among the communities in this setting.

## Supporting information

**S1 Data. Supplementary file field lab and malacology data.**
(XLSX)

## Acknowledgments

We thank the community health promoters who supported this study, Ministry of Health and Division of Vector Borne and Neglected Tropical Diseases (DVBNTD) both at the National and county Governments. We are very grateful to George Ogara, Kennedy Andiego, Meredith Odhiambo, Polycarp Odongo, Churchil Orao, John Oguso, Peter Olila, Austin Okinyo, Enock Sichangi, James Emisiko, Austin Wamalwa and Gabriel Wamalwa for supporting collection and identification of snails and who worked tirelessly to ensure the survey was a success. We also acknowledge the support from the Ministry of Interior and National Administration, and the National and County governments of Kakamega and Bungoma.

## Author contributions

**Conceptualization:** Maurice R Odiere, Jimmy Kihara, Dickson Kioko, Florence Wakesho, Dollycate N. Wanja, Ivy Sempele, Sultani Hadley Matendechero, Wyckliff Omondi.

**Data curation:** Maurice R Odiere, Dollycate N. Wanja.

**Formal analysis:** Maurice R Odiere, Dollycate N. Wanja.

**Funding acquisition:** Sultani Hadley Matendechero, Wyckliff Omondi.

**Investigation:** Jimmy Kihara.

**Methodology:** Stella Kepha.

**Project administration:** Maurice R Odiere, Martin Muchangi, Irene Chami.

**Resources:** Irene Chami.

**Supervision:** Stella Kepha, Jimmy Kihara, Chitiavi Juma, Dickson Kioko, Florence Wakesho.

**Writing – original draft:** Maurice R Odiere, Stella Kepha.

**Writing – review & editing:** Maurice R Odiere, Stella Kepha, Chitiavi Juma, Dickson Kioko, Florence Wakesho, Dollycate N. Wanja, Martin Muchangi, Ivy Sempele, Irene Chami, Sultani Hadley Matendechero, Wyckliff Omondi.

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
