## [Decision Letter · Decision Letter 0]

23 Dec 2025

PNTD-D-25-01648

Presence and infectivity of intermediate hosts of Schistosomiasis in Kakamega and Bungoma counties, western Kenya: Confirmation of autochthonous transmission

Dear Dr. Odiere,

Thank you for submitting your manuscript to PLOS Neglected Tropical Diseases. After careful consideration, we feel that it has merit but does not fully meet PLOS Neglected Tropical Diseases's publication criteria as it currently stands. Therefore, we invite you to submit a revised version of the manuscript that addresses the points raised during the review process.

Please submit your revised manuscript within by Feb 21 2026 11:59PM. If you will need more time than this to complete your revisions, please reply to this message or contact the journal office at plosntds@plos.org. Please include the following items when submitting your revised manuscript:

We look forward to receiving your revised manuscript.

Kind regards,

Jennifer A. Downs, M.D., Ph.D.

Academic Editor

Uriel Koziol

Section Editor

Shaden Kamhawi

co-Editor-in-Chief

Paul Brindley

co-Editor-in-Chief

Additional Editor Comments:

We appreciate the opportunity to review this important manuscript. We look forward to a revised version of this manuscript that will include more details, as outlined below.

Regarding one recommendation to provide confirmation using molecular identification of a subset of the collected snails, we recoganize that this may not be possible. If not possible, then this could be acknowledged as a limitation.

Journal Requirements:

At this stage, the following Authors/Authors require contributions: Maurice R Odiere. Please ensure that the full contributions of each author are acknowledged in the "Add/Edit/Remove Authors" section of our submission form.

Potential Copyright Issues:

- Please confirm (a) that you are the photographer of Figure 2., or (b) provide written permission from the photographer to publish the photo(s) under our CC BY 4.0 license.

- Figures 1 and 3. Please (a) provide a direct link to the base layer of the map (i.e., the country or region border shape) and ensure this is also included in the figure legend; and (b) provide a link to the terms of use / license information for the base layer image or shapefile. We cannot publish proprietary or copyrighted maps (e.g. Google Maps, Mapquest) and the terms of use for your map base layer must be compatible with our CC BY 4.0 license.

5) In the online submission form, you indicated that "Data will be made available on reasonable request". All PLOS journals now require all data underlying the findings described in their manuscript to be freely available to other researchers, either

1. In a public repository

2. Within the manuscript itself

3. Uploaded as supplementary information.

Reviewers' Comments:

Reviewer's Responses to Questions

Key Review Criteria Required for Acceptance?

Methods

-Are the objectives of the study clearly articulated with a clear testable hypothesis stated?

-Is the study design appropriate to address the stated objectives?

-Is the population clearly described and appropriate for the hypothesis being tested?

-Is the sample size sufficient to ensure adequate power to address the hypothesis being tested?

-Were correct statistical analysis used to support conclusions?

-Are there concerns about ethical or regulatory requirements being met?

Reviewer #1: See Summary and General Comments

Reviewer #2: 1. The objectives of the study were clearly stated but not hypothesis was not defined.

2. The methodology was detailed enough but no specific study design was stated.

3. The target population was well described and detailed for the methodology.

4. The sample size was purposive and large enough.

5. The statistical analysis used correctly support the conclusions.

6. Ethical approval was properly obtained.

Reviewer #3: The objectives of the study were clearly articulated and the methodology is okay.

Results

-Does the analysis presented match the analysis plan?

-Are the results clearly and completely presented?

-Are the figures (Tables, Images) of sufficient quality for clarity?

Reviewer #1: See Summary and General Comments

Reviewer #2: 1. The analysis presented tallies with the data analysis.

2. The results are clearly and completely presented.

3, The tables and figures are clear enough except for table 5.

Reviewer #3: The analysis presented matched the analysis plan, however the results need to be clearly presented in some sections.

Conclusions

-Are the conclusions supported by the data presented?

-Are the limitations of analysis clearly described?

-Do the authors discuss how these data can be helpful to advance our understanding of the topic under study?

-Is public health relevance addressed?

Reviewer #1: See Summary and General Comments

Reviewer #2: 1. The conclusion supports the data presented.

2. The limitations of the study were clearly stated.

3. The manuscript clearly explained and effectively related the usefulness of the data collected to better understand the topic. It also proffered recommendations for the advancement of the field of study.

4. The public health importance of the study was well stated and properly related.

Reviewer #3: The conclusions are relevant and the limitations of the study was clearly stated.

Editorial and Data Presentation Modifications?

Reviewer #1: See Summary and General Comments

Reviewer #2: Title: Kindly ensure that schistosomiasis is written in lowercase.

Schistosoma mansoni, Schistosoma haematobium and other scientific names included in the manuscript should be italicized.

Introduction:

The third paragraph, include reference (s) in “Three main endemic……………respectively”.

Materials and Methods

Kindly change the “C” in Kakamega County and Bungoma County to lowercase.

Snail sampling: Include the number of personnel per team.

Result

Kindly rephrase for the purpose of clarity “A total of 4,245 snails were collected across the 42 sites, with all but one yielding snails”.

Table: How the proportion of snails (%) was gotten is not clear, kindly include an explanation in statistical analysis.

Discussion

This point is not applicable to this study because a single season was observed – “Our findings corroborate the pattern observed for Bulinus spp. in Iringa, Tanzania, where Schistosoma spp. cercarial emergence was higher during the dry season [57],………… (presumably when temperatures are lower)”.

I suggest deleting this statement because it does not reflect the objective of the study, which is not season based – “Elsewhere, Gouvras and others [31] observed that shedding by Biomphalaria did not show any variation between dry and rainy seasons.

Reviewer #3: Minor Revision

Summary and General Comments

Reviewer #1: Reviewer summary

This manuscript presents a relevant and timely study addressing the distribution and infection status of intermediate snail hosts of Schistosoma in two counties of western Kenya, contributing valuable local-scale data to our understanding of transmission dynamics outside the Lake Victoria basin that is well described in other manuscripts. The paper is well written, with little to no grammatical errors, and generally well structured, and the fieldwork represents an applaudable and important effort in mapping potential foci of autochthonous transmission in areas where only human prevalence data is available. However, in its current form, the manuscript requires substantial revision before it can be considered for publication. Several essential aspects of data presentation, analytical clarity, and methodological transparency need improvement to ensure the results are interpretable, reproducible, and robust. My comments below are intended to be constructive and supportive of the authors’ work and provide guidance on how to strengthen the manuscript to meet the standards of PLOS Neglected Tropical Diseases.

General comments on the manuscript content

Only a suggestion – but the term snail ‘vectors’ is increasingly being used, and could be used throughout this manuscript (such as suggested in the title).

Cercariae shed from snails are being presumed as S. mansoni cercariae, can the authors convincingly rule out the possibility that cercariae could not be S. rodhaini in this locality? My knowledge of S. rodhaini distribution outside of the Lake Victoria shores is somewhat patchy (as is the literature), but after doing a bit of further research, what I can say is that in two publications from Saoud (1966) - https://www.cambridge.org/core/journals/journal-of-helminthology/article/abs/on-the-morphology-of-schistosoma-rodhaini-from-kenya/462782B093093D8CA10BE7921BEF51F2 - AND - https://www.cambridge.org/core/journals/journal-of-helminthology/article/abs/susceptibility-of-some-planorbid-snails-to-infection-with-schistosoma-rodhaini-from-kenya/CE0289C35F756E7F7B3A998613B3A8E4 they reference work by Nelson (1962) in chapter 7 of the Ciba Foundation Symposium ‐ Bilharziasis (a piece of work I cannot get hold of unfortunately) that they successfully infected Biomphalaria pfeifferi nairobiensis (now synonymised to B. pfeifferi) with S. rodhaini isolates from Kisumu region. Saoud then in the second publication described the use of this S. rodhaini isolate to infect snails in their lab, and also got a 37% infection rate of exposed B. pfeifferi. See also Brown (1994) Freshwater Snails of Africa – under B. pfeifferi description for further information and references on this species compatability with mammalian schistosomes. All in all, I think here it would be great to mention that some of these untyped schistosome infections may be S. rodhaini and not S. mansoni, but that is still to be established in future studies. The authors could even conduct schistosome targeted PCRs from snail tissue extracts of the samples collected, and therefore still type species given that it seems as though cercariae were not collected.

“Infectivity” is used throughout the manuscript to refer to the number of snails found infected. I think this terminology should be changed to an alternative since the definition of infectivity, see: https://en.wikipedia.org/wiki/Infectivity

“In epidemiology, infectivity is the ability of a pathogen to establish an infection. More specifically, infectivity is the extent to which the pathogen can enter, survive, and multiply in a host. It is measured by the ratio of the number of people who become infected to the total number exposed to the pathogen.”

To apply this term to snails it would mean to say that one is testing the number of snails that get infected following exposure, but the snails collected in this study may not necessarily have been exposed before. It would be better to just use the commonly used term of ‘prevalence’ when discussing the number of snails found infected with parasites.

I mention this multiple times in the line by line comments below, but not in all instances. The word site/school is used interchangeably throughout the manuscript to represent the area surveyed for snail sampling sites. This makes the manuscript quite complicated to follow since sometimes the snail sampling sites are also referred to as sites, or points, or human water contact sites… Please ensure consistency and help the reader to follow. I would suggest calling the area sampled around the school that prevalence data is recorded from a previous study the ‘school region’, and then each snail sampling site a ‘human-water contact site’ or ‘snail sampling site’.

A supplementary dataset needs to be provided with this study that includes all sampling data per snail sampling site. This should include all variables recorded (from GPS of human water contact points, snail species and abundance of each found, to the human activities reported at each). This will make the study more citable, reproducible and the data available for others to use in meta-analysis and such, or when considering transmission control in the area. Providing the full dataset broken down by each site would then allow for the ‘Availability of data and materials’ statement to ring true.

The methodology of how many sites were sampled per school region is not sufficient to make this study reproducible. The current explanation given is:

“In each site/village, water bodies including canals, rivers, streams, ponds, swamps/marshes and dams (Fig. 2) where there was evidence of human-water contact activities were selected for snail sampling with the help of local community health promoters (CHPs).”

However, there is no measure on how far away from the schools, sites are sampled, or how many sites per region? Was a radius around the school chosen for sampling within? Was it time limited per day which dictated how many sites could be sampled per school? Please clarify this to establish how extensively regions were sampled.

Line by line comments

Title

The title is somewhat misleading in my opinion.

1. Snails are intermediate hosts of the parasite Schistosoma spp., not the disease schistosomiasis.

2. The term infectivity, to me, seems to suggest that the study is practically testing the infectivity / compatibility of snail hosts to schistosomes, but actually, the study is measuring the prevalence of schistosome infected snails collected.

3. Although the study does find Bulinus globosus, it does not find any infected snails or relate back to human data on Sh. I suggest the title therefore more directly refers to the Biomphalaria pfeifferi species which is associated with the infected snail findings and past human prevalence data.

4. As per my general comment above, as no molecular data truly confirms the cercs are S. mansoni, can one be confident to use the species name in the title?

I would suggest a title change to something more relevant to the study, like:

Biomphalaria pfeifferi infected with Schistosoma spp. in Kakamega and Bungoma counties, western Kenya, implies autochthonous transmission of Schistosoma mansoni

Abstract

Clarify what the evidence is, can one simply not just say ‘found individuals infected with Schistosoma mansoni in this area’

Change to ‘follow-up malacological survey’

Change infectivity to ‘prevalence of S. mansoni infections in snails was determined.’

Specify in abstract that each waterbody was surveyed one time during the study.

It seems unusual to me to display confidence intervals on the total abundance of snails identified to each species (and number infected), since this is a direct measure of quantity rather than demonstrating variability around an estimate (like a mean). For the mean abundance – a range is given here as a presumably a standard deviation? Please correct me if I am wrong here.

It would be useful to know from the abstract at how many sites schistosome infected snails were found, and how widely spread across the study population. How about where infected snails were most likely to be found in terms of waterbody type? This all seems like information very relevant to the study that is not provided in the abstract.

Introduction

In the last sentence of the first paragraph:

“The existence of persistent hotspots of schistosomiasis transmission [8, 9] underscore the challenge of spatial heterogeneity in MDA success which is likely multifactorial, with snail distribution and abundance predicted to play a large role [11, 12].”

One could also mention the role of snail genetics associated with immunity and compatibility of the snails with S. mansoni, which could go some way to explaining presence of persistent hotspots. There are recent papers discussing exactly this on B. sudanica in Lake Victoria.

“the type of snail species “ – there is not a ‘type’ of snail species per se, I would change to just “Delimiting the species of snail present..”

“and the Lake Victoria basin area (mainly S. mansoni)..” – not clearly what ‘mainly’ refers to. Maybe state “S. mansoni with focal pockets of S. haematobium transmission” with a reference for clarity?

“with a dearth of surveys in areas further west” – maybe make clear that by further west you mean western Kenya and not further west of Lake Victoria (which would be Uganda of course!). Maybe just state that dearth of surveys in counties of Western Kenya that do not border the lake shore?

“For a long time, it was widely believed that there was no or very little schistosomiasis in the four counties of western Kenya (Kakamega, Bungoma, Vihiga and Trans Nzoia)” – State when ‘a long time’ was exactly - what period were surveys conducted in the past before the data in 2021 was added?

“the presence, species type” – again there is no ‘type’ of species, it is just a species or not. Please check this terminology throughout and correct.

Materials and Methods

Study area and population

The Figure 1 map could be improved if possible, showing neighbouring countries and regions other than Kenya for one, giving the sites relative position to Lake Victoria in the larger scale map would be nice to provide context for readers unfamiliar with the area. Also can the Yala and Isiukhu rivers be shown on there? There are many freely available freshwater body ‘layers’ / vectors available online that could be used (like shown in Figure 3!). Other features described in this paragraph would be really nice to be shown on the map too. other rivers, dams etc… Could the map also show ward boundaries since these are later discussed?

The picture is also quite pixelated in my reviewer copy, but this may be due to exporting and formatting. If difficult in R, I would suggest using freely available GIS software such as QGIS to make maps. Please improve in the revised copy.

For estimated population size, say what year this is for.

Could the map also show relevant to the previous granular mapping work where human Schistosoma spp. infections are found? This would be helpful for putting the snail collections and gastropod infections in context.

Why have the authors chosen to interchangeably use schools / sites to mean the same thing? This is somewhat confusing since snail localities are often referred to as ‘sites’.

Study sites sampled

Text on site selection a bit confusing, suggestions:

“10 Wards with 22 sites were included in Kakamega and 9 Wards with 20

sites for Bungoma” – combine with the first sentence of paragraph when mentioned 19 sites selected.

“The selected sites were among those included in the granular mapping…” – You already mentioned this in first sentence of the paragraph – combine these sentences for clarity.

“Each of the 42 sites was visited once during the dry season (February 2025).” – This is where it gets confusing with the schools / sites definition. You visited multiple human water contact sites (rivers / streams) per ‘site / school’. This sentence sounds like just 42 freshwater sites were visited. How many freshwater sites were actually surveyed across the 42 regions surrounding the schools selected I presume will come up in the results.

Also – how far outside the school region were sites sampled? Was a radius site around the school to determine how many sites sampled? Was every site within the ward sampled?? Was it limited by time to survey within a day to do as many as possible per school region? Currently this is not clear and the sampling strategy of how snail sites were identified and selected needs to be explained better.

Can the sites in Figure 2 be linked directly to GPS coordinates or site names from the surveys? Is there a supplementary dataset that lists the exact GPS coordinates that were recorded at each snail sampling site and the snails collected from there? This data would be very useful for future reference.

Snail screening for cercariae

Were any of the cercariae released from snails preserved in ethanol or another DNA capture media?

Statistical analysis

The ‘sites’ and ‘snail sampling localities’ is confusing in this sentence also:

“The mean B. pfeifferi snail abundance was computed as the total number of snails collected per site divided by the number of points sampled per site.” – so snail collecting sites now called points?

and also here…: “number of snails collected at a site with the initial prevalence in SAC at the corresponding site”

Results

Snail abundance

“A total of 4,245 snails were collected across the 42 sites” – but how many human water contact sites for snail collection surveyed?? This is more relevant to report in relation to the abundance.

“rich snail species diversity” – rich compared to what? Remove descriptives in results and save for discussion when comparing and contrasting to other studies.

“7 freshwater snails…” – change to “7 freshwater snail species”

“Not surprisingly, we did not find any Biomphalaria sudanica, a species mainly found and involved in transmission around Lake Victoria [19, 31].” – this sentence is for the discussion and not results.

Throughout the results discussing the sites compare to human prevalence – it is not clear how close these sites are to the schools surveyed or inhabitants infected with S. mansoni.

How about providing results on the abundance of snails / species relevant to the type of water body sampled? This seems more biologically relevant rather than comparing snail abundances between county boundaries. EDIT: can see that this is brought in in a later section – it would make sense to include it all here in the snail abundance section.

Table 1 – as mentioned earlier, it seems unusual to report 95% CI with overall abundance values but correct me if I am wrong.

Table 2 – could table 2 also include the number of snails that shed mammalian schistosome cercariae rather than any type of cercariae per ward. This is more relevant for the paper. The following table 3 is much bigger and therefore harder to determine exactly where the majorities of mammalian schisto infections were observed.

Spatial distribution of snails and Schistosoma cercarial shedding

Table 3 – it seems key to add in here how many snail sampling sites were actually surveyed per ‘school’. This will have an impact on the snail abundance and number infected? Although given by ward in Table 2, it should be broken down further here by locality.

Is B. pfeifferi abundance noted in sites in the Nzoia or tributaries or? From Figure 3 map, it looks like many sites further away from Nzoia that have an abundance of Bp too? So what does this mean?

“The highest number of B. pfeifferi (605 snails) was recorded at Indangalasia site (Table 3)…” – but again is this a reflection on there being a higher number of snail collection sites that were surveyed in that region? The summaries of abundance per school region could be misleading if not put in context of the sampling effort at each one.

Fig 3 – the figure caption for Figure 3 is somewhat misleading – since the 42 points is per school, and the abundance of Biomphalaria is summed from all the sites withing that region associated with the school – correct? So therefore “…abundance of Biomphalaria snails at each of the sampled points…” seems to suggest that there is a point on the figure per snail sampling site – but this is actually school locality.

“Prevalence was highest at the Kisembe primary..” – the wording of this sounds like talking about human prevalence but refers to snails. Make clear when referring to snail infection prevalence and human prevalence. Also perhaps state in text here that the 72% was from 25 snails as shown in Table 3. Note this is a really high number of snails found naturally infected from such a small number, and quite unusual. Please discuss such things in the discussion. Were snail rinsed before plating to avoid cross contamination from the pot snails grouped in from the collection site?

Table 4 – may as well add schistosome cercariae to this table

Distribution of snails of Biomphalaria pfeifferi snails among water bodies in Kakamega and Bungoma counties

The reporting of the results could be improved and made easier to follow. It is clear that most snails were collected from streams, reporting as the ‘proportion by water body type’ is somewhat confusing and non-sensical if you just sampled from more stream sites than any other and therefore this represent the majority of your collection (as clear in table 5).

“The mean B. pfeifferi abundance at the streams was higher compared to rivers (P = 0.005).” – is it not significantly higher than ponds also? Was highest mean abundance from swamps / marshes when counting all sites? The text really doesn’t reflect much of what is presented in the table leaving the reader to interpret themselves.

Table 5 – The lowercase letters representing significant differences don’t make sense too – so for ‘a’ – streams swamps and marshes are all significantly different to each other? Based on what statistical test? This should be detailed in the table caption even if elsewhere in the methods as all tables and figures should be interpretable as stand-alone items.

Final column of table 5 should also be the ‘Total number of snails… in each water body type’

Human water contact activities

It is a missed opportunity here to not perform any analysis connecting human behaviour to snail presence / absence and infected Biomphalaria with schisto snail presence. Is this data stored and reported elsewhere? This should be in the supplementary dataset.

Discussion

The discussion is well written, however I expected it will be revised in a the next version of the manuscript following the comments above.

Reviewer #2: The study is a very important one because it will add to the available knowledge that underscores the need to integrate snail control in tackling schistosomiasis, particularly in endemic developing countries. However, the credibility of the report would have been stronger if confirmation using molecular identification of a subset of the collected snails was carried out, rather than solely relying on putative morphological identification that is not reliable.

Reviewer #3: Presence and infectivity of intermediate hosts of Schistosomiasis in Kakamega and Bungoma counties, western Kenya: Confirmation of autochthonous transmission

This study was conducted to ascertain the presence of Biomphalaria snail intermediate hosts following evidence of intestinal schistosomiasis infections in the area. The study is important for understanding the transmission of schistosomiasis in the region. The manuscript is fairly well written and would benefit from improvements.

The authors should conduct grammar formatting to improve the flow of the manuscript. Repetitions should be avoided to make it concise.

Abstract

1. Nineteen Wards which had Schistosoma mansoni prevalence ≥10% were selected from Kakamega and Bungoma counties, from which 42 sites (schools) that had the highest prevalence of S. mansoni were used to identify proximal water bodies with human-water contact activities to sample for intermediate hosts

Modify statement-long complex statement. Meaning unclear. Break into two clear statements.

Introduction

1. Whereas chemotherapy

Change to Though chemotherapy

Methods

1. The study involved 42 sites (schools) purposively selected from 19 Wards with the highest prevalence of S. mansoni in school-age children (SAC) in Kakamega and Bungoma counties (Fig. 2)

Fig 2 is not correct.

2. (range: 1-7)

It is not clear what this means.

Results

1. Whereas the mean B. pfeifferi abundance was lower in sites with ≥10% S. mansoni prevalence

Remove “whereas”

2. For instance, at Okanya site in Bungoma, one of the sites with <10% S. mansoni prevalence it was high (115 snails),

Restructure statement. What do you mean by site in Okanya site? Are yiu referring to a type of water body, a river, swamp etc?

3. Whereas the mean B. pfeifferi abundance was lower in sites with ≥10% S. mansoni prevalence compared to sites with <10% S. mansoni prevalence (Y = -1.54x + 3.91, P = 0.011), it was also highly variable among sites. For instance, at Okanya site in Bungoma, one of the sites with <10% S. mansoni prevalence it was high (115 snails), while the mean abundance was low (1 snail) in Mahola site in Kakamega, one of the sites with ≥10% S. mansoni prevalence.

Where is the data supporting this statement?

4. Elaborate on Table 2

5. Spatial distribution of snails and Schistosoma cercarial shedding

The authors should remove the column “No. of B. globosus”, the No. of B. pfeifferi (+) for S. mansoni should be combined with that of “Prev of S. mansoni in B. pfeifferi” as n (%). of S. mansoni (+) B. pfeifferi, the CI limit should be placed in a single column and written as xx-yy. The discussion should centre on Biomphalaria. The data for B. globosus can be placed in a new table with data on cercaria shedding as well.

6. Human-water contact activities Several human-water contact activities were observed and noted among the different water bodies sampled. The activities included swimming, bathing, and fetching water for drinking and laundry. Other observed activities were irrigation, fishing and sand harvesting.

Remove this section

Discussion

7. The reasons for absence of snails in some regions should be discussed.

General comments

Conduct grammar formatting. There are many ambiguous statements in the manuscript.

PLOS authors have the option to publish the peer review history of their article (what does this mean?). If published, this will include your full peer review and any attached files.

Do you want your identity to be public for this peer review? For information about this choice, including consent withdrawal, please see our Privacy Policy.

Reviewer #1: Yes: Tom Pennance

Reviewer #2: No

Reviewer #3: No

Figure resubmission: While revising your submission, we strongly recommend that you use PLOS’s NAAS tool (https://ngplosjournals.pagemajik.ai/artanalysis) to test your figure files. NAAS can convert your figure files to the TIFF file type and meet basic requirements (such as print size, resolution), or provide you with a report on issues that do not meet our requirements and that NAAS cannot fix. 
---

## [Decision Letter · Decision Letter 1]

31 Mar 2026

PNTD-D-25-01648R1Biomphalaria pfeifferi infected with Schistosoma spp. in Kakamega and Bungoma counties, western Kenya confirms autochthonous transmission of Schistosoma mansoniPLOS Neglected Tropical Diseases Dear Dr. Odiere, Thank you for submitting your manuscript to PLOS Neglected Tropical Diseases. After careful consideration, we feel that it has merit but does not fully meet PLOS Neglected Tropical Diseases's publication criteria as it currently stands. Therefore, we invite you to submit a revised version of the manuscript that addresses the points raised during the review process. Please submit your revised manuscript by Apr 30 2026 11:59PM. If you will need more time than this to complete your revisions, please reply to this message or contact the journal office at plosntds@plos.org.  Please include the following items when submitting your revised manuscript:* A letter that responds to each point raised by the editor and reviewer(s). You should upload this letter as a separate file labeled 'Response to Reviewers'. This file does not need to include responses to any formatting updates and technical items listed in the 'Journal Requirements' section below.* A marked-up copy of your manuscript that highlights changes made to the original version. You should upload this as a separate file labeled 'Revised Manuscript with Track Changes'.* An unmarked version of your revised paper without tracked changes. You should upload this as a separate file labeled 'Manuscript'. If you would like to make changes to your financial disclosure, competing interests statement, or data availability statement, please make these updates within the submission form at the time of resubmission. Guidelines for resubmitting your figure files are available below the reviewer comments at the end of this letter. We look forward to receiving your revised manuscript. Kind regards, Jennifer A. Downs, M.D., Ph.D.Academic EditorPLOS Neglected Tropical Diseases Uriel KoziolSection EditorPLOS Neglected Tropical Diseases

Shaden Kamhawi

co-Editor-in-Chief

Paul Brindley

co-Editor-in-Chief

 Additional Editor Comments: We appreciate the authors' attention to detail in this revised version of the manuscript. The remaining comment from the reviewer is important, and we request the authors to clarify in the Methods section that the cercariae species cannot be distinguished morphologically between S. mansoni and S. rodhaini, and to discuss this more clearly in the limitations of the discussion section.

Please also ensure that the data sets are included with the next submission. They are not currently visible with the revised submission.  Journal Requirements:

Please provide separate figure files in .tif or .eps format. For more information about figure files please see our guidelines:

https://journals.plos.org/plosntds/s/figures#loc-file-requirements

 Reviewers' comments: Reviewer's Responses to Questions

Key Review Criteria Required for Acceptance?

Methods:

-Are the objectives of the study clearly articulated with a clear testable hypothesis stated?

-Is the study design appropriate to address the stated objectives?

-Is the population clearly described and appropriate for the hypothesis being tested?

-Is the sample size sufficient to ensure adequate power to address the hypothesis being tested?

-Were correct statistical analysis used to support conclusions?

-Are there concerns about ethical or regulatory requirements being met?

Reviewer #1: See Summary and General Comments.

Results

-Does the analysis presented match the analysis plan?

-Are the results clearly and completely presented?

-Are the figures (Tables, Images) of sufficient quality for clarity?

Reviewer #1: See Summary and General Comments.

Conclusions

-Are the conclusions supported by the data presented?

-Are the limitations of analysis clearly described?

-Do the authors discuss how these data can be helpful to advance our understanding of the topic under study?

-Is public health relevance addressed?

Reviewer #1: See Summary and General Comments.

Editorial and Data Presentation Modifications?

Reviewer #1: See Summary and General Comments.

Summary and General Comments

Reviewer #1: Thank you to Odiere and colleagues for submitting the revised manuscript with accompanying comments on the revisions made. I am very happy with the changes made across the manuscript and would like to thank the authors for taking their time to do this. I am recommending that the manuscript be accepted, however I am submitting as minor revision needed I would like to ask that the authors reconsider my point on cercariae ID as I have detailed again further AND I also did not receive a supplementary dataset with this submission that was stated by the authors (see below).

Cercariae identification

I would like to reiterate a point from my initial comments, which I do still believe could be addressed better, regarding the identification of schistosome cercariae to the species level. In the rebuttal, the authors states that cercariae morphology is sufficient to identify schistosome cercariae, and also state this again in the revised discussion, however the guides used by Frandsen as cited by the authors, only allow for an identification of cercariae to the genus Schistosoma, and not the species level as cercariae morphology overlaps. See this direct quote from Frandsen page 199:

“Numerous species of the genus Schistosoma exist in Africa (see Table 2) and assignment of a Schistosoma sp. cercaria to the species level is for the non-specialist not possible on the basis of morphological criteria because morphological differences between the cercariae of the different species are extremely small.”

Therefore, all schistosome cercariae morphological identified using this guide (which the authors used) should be listed as Schistosoma sp.. In the discussion, one can then make the point that it is likely that these are S. mansoni rather than S. rodhaini based on the associated epidemiology and, to some extent, the shedding pattern... On that note, I do agree with the authors that the shedding patterns and timings of cercariae emergence are a great proxy to make a better guess of schistosome species ID, and I would suggest that the authors please edit the methods section “Snail screening for cercariae” with appropriate citations (as given in the discussion) that support the timing of shedding the snails in the methods, even though the time the authors shed snails is not necessarily optimal for S. mansoni (as the authors discuss in the limitations section of the discussion), it is notable also that the time they shed the snails overlaps with that of S. rodhaini in previous studies, for example see:

Norton, A., Rollinson, D., Richards, L. et al. Simultaneous infection of Schistosoma mansoni and S. rodhaini in Biomphalaria glabrata: impact on chronobiology and cercarial behaviour. Parasites Vectors 1, 43 (2008). https://doi.org/10.1186/1756-3305-1-43

“ S. rodhaini's entire shedding period extended from 1430–1030 hours, however shedding was at a low level (<50 cercariae per snail) for all apart from the peak 2 hour period (1830–2030 hours).”

The authors state in their ms that the shed from 14-16:00 hours, which overlaps with this period. I would suggest that this is specifically stated by the authors, and that at least some mention to the S. rodhaini species is mentioned in this manuscript, as it is currently not mentioned at all and I think doing so only strengthens this manuscript.

Overall, this will be a small edit in the methods as suggested above, changing S. mansoni to ‘Schistosoma sp. cercariae’ in the results, and then in the discussion bring together to discuss the shedding Biomphalaria as being S. mansoni.

Supplementary material

I would also like to make clear that the supplementary material/dataset was not made available to me while performing this second round of review, and do not see it cited within the manuscript anywhere, and I also note that in the final part of the manuscript it states:

“Availability of data and materials

All data supporting the findings of this study are available within the manuscript.”

Which is now not the case if supplementary data is also included... I hope that this dataset can be included in the final manuscript as it will be highly valuable.

PLOS authors have the option to publish the peer review history of their article (what does this mean?). If published, this will include your full peer review and any attached files.

Do you want your identity to be public for this peer review? For information about this choice, including consent withdrawal, please see our Privacy Policy.

Reviewer #1: Yes: Tom Pennance

  Figure resubmission: While revising your submission, we strongly recommend that you use PLOS’s NAAS tool (https://ngplosjournals.pagemajik.ai/artanalysis) to test your figure files. NAAS can convert your figure files to the TIFF file type and meet basic requirements (such as print size, resolution), or provide you with a report on issues that do not meet our requirements and that NAAS cannot fix.

After uploading your figures to PLOS’s NAAS tool - https://ngplosjournals.pagemajik.ai/artanalysis, NAAS will process the files provided and display the results in the "Uploaded Files" section of the page as the processing is complete. If the uploaded figures meet our requirements (or NAAS is able to fix the files to meet our requirements), the figure will be marked as "fixed" above. If NAAS is unable to fix the files, a red "failed" label will appear above. When NAAS has confirmed that the figure files meet our requirements, please download the file via the download option, and include these NAAS processed figure files when submitting your revised manuscript. Reproducibility: To enhance the reproducibility of your results, we recommend that authors of applicable studies deposit laboratory protocols in protocols.io, where a protocol can be assigned its own identifier (DOI) such that it can be cited independently in the future. Additionally, PLOS ONE offers an option to publish peer-reviewed clinical study protocols. Read more information on sharing protocols at https://plos.org/protocols?utm_medium=editorial-email&utm_source=authorletters&utm_campaign=protocols

---

## [Editor Report · Decision Letter 2]

16 Apr 2026

Dear Dr Odiere,

We are pleased to inform you that your manuscript 'Biomphalaria pfeifferi infected with Schistosoma spp. in Kakamega and Bungoma counties, western Kenya confirms autochthonous transmission of intestinal schistosomiasis ' has been provisionally accepted for publication in PLOS Neglected Tropical Diseases.

Best regards,

Jennifer A. Downs, M.D., Ph.D.

Academic Editor

Uriel Koziol

Section Editor

Shaden Kamhawi

co-Editor-in-Chief

Paul Brindley

co-Editor-in-Chief

---

## [Editor Report · Acceptance letter]

Dear Dr Odiere,

We are delighted to inform you that your manuscript, "Biomphalaria pfeifferi infected with Schistosoma spp. in Kakamega and Bungoma counties, western Kenya confirms autochthonous transmission of intestinal schistosomiasis ," has been formally accepted for publication in PLOS Neglected Tropical Diseases.

Best regards,

Shaden Kamhawi

co-Editor-in-Chief

Paul Brindley

co-Editor-in-Chief
